

# Evaluating the Turkish validity and reliability of the Brief Illness Perception Questionnaire in periodontal diseases

Canan Önder[1] and Batuhan Bakirarar[2]

[1] Department of Periodontology, Faculty of Dentistry, Ankara University, Ankara, Turkey
[2] Department of Biostatistics, Faculty of Medicine, Ankara University, Ankara, Turkey

## ABSTRACT

**Background:** The Brief Illness Perception Questionnaire (Brief IPQ) is a widely used multifactorial scale that assesses the individuals' perceptions of illness. Although there are studies investigating the psychometric properties of the Brief IPQ in many languages, the Turkish version of Brief IPQ on periodontal diseases has not been revealed so far. This study aimed to evaluate the Turkish validity and reliability of the Brief IPQ and contribute to the literature. It is also aimed to evaluate the patients' illness perception with periodontal disease and to reveal the possible effects of the disease on the patients' daily life.

**Methods:** This cross-sectional study was conducted with 209 patients with periodontal diseases (137 gingivitis and 72 periodontitis cases). Sociodemographic characteristics and clinical periodontal measurements of all patients were recorded. The Turkish versions of the Brief IPQ and the HAD Scale were applied to the patients *via* face-to-face. The construct validity was determined using confirmatory factor analysis. Test–retest reliability and internal consistency were performed using ICC test and Cronbach's alpha, respectively. The concurrent validity was determined by using Spearman's correlation coefficient.

**Results:** The confirmatory factor analysis showed that the scale has one factor. The Spearman's correlation coefficient results were found 0.843 and 0.854 for concurrent validity. Cronbach's alpha value of the scale was 0.944 in the internal consistency analysis. ICC value was found to be 0.987 for test-retest reliability. Floor/ceiling effects were considered not to be present.

**Conclusions:** It was found that the Turkish version of The Brief Illness Perception Questionnaire is valid and reliable. Brief IPQ may be used to determine the illness perception in patients with periodontal diseases.

Corresponding author
Batuhan Bakirarar,
batuhan_bakirarar@hotmail.com

## INTRODUCTION

The study of individuals' perceptions of illness is based on researchs in the 1960s (*Petrie & Weinman, 1997*; *Petrie et al., 2002*). Early research identified five dimensions within the cognitive perception of illness: identity (the words that the patient uses to describe the illness, and the symptoms that the patient believes as part of the illness), consequences (expected effects and outcomes of illness), cause (personal ideas about the cause of illness),

timeline (the belief of the patient that how long illness will last) and cure or control (patient's belief about recover from or control the illness) (*Lau & Hartman, 1983*; *Leventhal, Nerenz & Steele, 1984*).

It is important in all areas to consider patients' perceptions of illness as part of the psychosocial assessment. Improving patient awareness of disease perceptions can improve treatment outcomes and physicians' communication with patients. This awareness has revealed the effect of psychometric parameters on the natural course of chronic diseases in the last decade (*Ferreira et al., 2017*). Over the years, treatment success has evolved into a term that encompasses the individual's behavior and various social, psychological and emotional aspects in addition to clinical recovery (*Mariotti & Hefti, 2015*; *Khan et al., 2021*). In periodontal disease, which is a chronic inflammatory disease, the development of patients' perception of the disease and understanding their psychological characteristics can help clinicians design a successful periodontal treatment plan. It has been shown that periodontal diseases play an important role on the patient's quality of life and this effect becomes more pronounced as the severity or prevalence of the disease increases (*Buset et al., 2016*). Therefore, in the treatment process of periodontal diseases, not only the plaque level, but also all factors that can affect the onset of the disease should be taken into account with a holistic approach.

Early studies investigating the content of perceptions of illness included mostly open-ended questions. Since it was believed that the information obtained from open-ended questions was immeasurable, scales were started to be developed. The Illness Perception Questionnaire (IPQ) is a widely used multifactorial scale that assesses five representations of cognitive illness on a five-point Likert scale and consists of 38 items (*Weinman et al., 1996*). In a revised version of this scale, Illness Perception Questionnaire-Revised (IPQ-R), 48 items were added to the original scale to create new subscales (*Moss-Morris et al., 2010*).

The IPQ is a tool used to apply Leventhal's self-regulation model in the clinical setting (*Weinman et al., 1996*). Since the IPQ and IPQ-R consist of approximately 80 items, a simpler and shorter version, Brief IPQ, was developed. The use of this scale, is particularly useful in individuals with limited time for assessment, such as the very ill or the elderly, or in patient groups where repeated measurements are taken (*Broadbent et al., 2006*).

The original study demonstrated that the Brief IPQ showed good psychometric properties, including concurrent, predictive and discriminant validity. The discriminant validity of the scale was supported by its ability to distinguish different diseases such as asthma, diabetes, cold, myocardial infarction and chest pain (*Broadbent et al., 2015*). The Brief IPQ was also evaluated on periodontal diseases for patients' perception of illness (*Machado et al., 2019*; *Machado et al., 2020*; *Discepoli et al., 2022*). However, there is only one study evaluating the psychometric properties of the scale on periodontal diseases. *Machado et al. (2019)* applied the Portuguese version of the Brief IPQ to patients with gingivitis and periodontitis and reported that the scale showed acceptable reliability and construct factorial validity.

Brief IPQ has been translated into 26 languages, but the researches which were examining the construct validity of the scale is limited. Therefore, it is recommended that

the validity and reliability of the scale will be further evaluated in various populations (*Clark & Watson, 2019*; *Wasserman & Bracken, 2013*).

In the literature, there are limited number of scales that measure the level of disease perception of patients. This study aimed to evaluate the Turkish validity and reliability of the Brief IPQ and contribute to the literature. It is also aimed to evaluate the patients' illness perception with periodontal disease and to reveal the possible effects of the disease on the patients' daily life.

## MATERIALS AND METHODS

Portions of this text were previously published as part of a preprint (*Ersü, Bakırarar & Tatlı, 2023*).

### Study population and study design

This cross-sectional study included 209 patients who applied to Ankara University Faculty of Dentistry Department of Periodontology for periodontal treatment between December 2022 and March 2023. The research was carried out in accordance with the Declaration of Helsinki of 1975, as updated in 2000, and the study design was authorized by the Ankara University Faculty of Dentistry Clinical Studies Ethics Committee (ethical approval number: 36290600/68/2022). All patients included in the study provided their consent after being fully informed and written informed consent from was received from the participants. The patients who was ≥18 years old, who accepted to take part in voluntarily and individuals with the ability and willingness to give informed consent and understand the meaning of the questionnaire were included the study.

### Translation and cultural adaptation

For the validity and reliability study of The Brief Illness Perception Questionnaire (Brief IPQ), we contacted with Dr. Weinman *via* e-mail and we recieved required permission for using the scale in the present study.

Turkish adaptation of Brief IPQ was comprised of the following stages. First, the survey was translated to Turkish by five subject-matter experts. Next, the Turkish forms were back-translated to English to review the consistency between the two forms. The same experts discussed the translated Turkish forms and made the required semantic and grammatical corrections to finalize the Turkish form (Table 1).

### Work Items

#### The brief illness perception questionnaire

Brief IPQ was developed by *Broadbent et al. (2006)* and aimed to create a measure with an alternative form to the multi-factor Likert scale approach used in Illness Perception Questionnaire (IPQ) and Illness Perception Questionnaire Revised (IPQ-R). Brief IPQ was created by shortening the questions of IPQ-R. The Brief IPQ consists of eight items scored on an 11-point Likert scale, items scores range between 0–10. Each item reflects one of the following meanings: consequences, timeline, personal control, treatment control, identity, concern, understanding and emotional response. The ninth question is open-ended and asks the patients to list the three most important causes of their diseases. After the

**Table 1 Original and Turkish versions of the Brief IPQ.**

| Brief IPQ questions | Original[a] | Turkish |
|---|---|---|
| Consequences (item 1) | How much does your illness affect your life? | Hastalığınız hayatınızı ne kadar etkiliyor? |
| Timeline (item 2) | How long do you think your illness will continue? | Sizce hastalığınız ne kadar devam edecek? |
| Personal control (item 3) | How much control do you feel you have over your illness? | Hastalığınız üzerinde ne kadar kontrole sahip olduğunuzu hissediyorsunuz? |
| Treatment control (item 4) | How much do you think your treatment can help your illness? | Tedavinizin hastalığınıza ne kadar yardımcı olabileceğini düşünüyorsunuz? |
| Identity (item 5) | How much do you experience symptoms from your illness? | Hastalığınızın belirtilerini ne kadar hissediyorsunuz? |
| Concern (item 6) | How concerned are you about your illness? | Hastalığınız hakkında ne kadar endişelisiniz? |
| Understanding (item 7) | How well do you feel you understand your illness? | Hastalığınızı ne kadar iyi anlayabildiğinizi düşünüyorsunuz? |
| Emotional response (item 8) | How much does your illness affect you emotionally? (*e.g.*, does it make you angry, scared, upset or depressed?) | Hastalığınız sizi duygusal olarak ne kadar etkiliyor? (Örneğin, sizi kızdırıyor mu, korkutuyor mu, üzüyor mu ya da depresyona mı sokuyor?) |
| Three main causal factors in their illness (item 9) | Please list in rank-order the three most important factors that you believe caused your illness. The most important causes for me: | Lütfen hastalığınıza neden olduğuna inandığınız en önemli üç faktörü sıralayınız. Benim için en önemli sebepler: |

**Note:**
[a] *Broadbent et al. (2006)*.

questions 3, 4 and 7 are reverse coded, a Brief IPQ total score generated by summing up the eight questions. An increase in the total Brief IPQ score means that the disease is more threatening (*Broadbent et al., 2006*).

### Hospital anxiety and depression scale

The Hospital Anxiety and Depression (HAD) Scale which was developed by *Zigmond & Snaith (1983)* and of which Turkish validity and reliability study was conducted by *Aydemir et al. (1997)* was utilized as the reference scale in the present study. This scale consists of 14 items and two subscales in total which are Anxiety and Depression. Items 1, 3, 5, 7, 9, 11 and 13 are rated within Anxiety subscale, and higher scores from this subscale represent higher Anxiety. Items 2, 4, 6, 8, 10, 12 and 14 are rated within Depression subscale, and higher scores from this subscale represent higher Depression (*Aydemir et al., 1997*).

### Data collection

Sociodemographic characteristics and clinical periodontal measurements of all patients were recorded. The Turkish versions of the Brief IPQ and the HAD Scale were applied to 209 patients *via* face-to-face by a single trained periodontist (C.Ö.). After 4 weeks, the

scales were re-administered to the patients and 117 of 209 patients completed the questionnaires.

### Periodontal examination and diagnosis

The periodontal examination was performed by a single experienced researcher using a Williams periodontal probe. A total of 137 gingivitis and 72 periodontitis patients were included in this study. Clinical and radiological examinations of all participants were performed for diagnosis of gingivitis and periodontitis. The same investigator (C.Ö.) measured six regions of all teeth (mesiobuccal, buccal, distobuccal, mesiolingual, lingual and distolingual) to record the clinical periodontal measurements, including probing pocket depth (PPD), attachment loss (AL), and bleeding on probing (BOP). Gingivitis cases were defined according to *Trombelli et al. (2018)* and periodontitis was defined according to *Tonetti, Greenwell & Kornman (2018)*.

### Sample size

One of the suggested approaches for sample size calculation in scale development studies is to include patients with 20 times the number of items in the scale (*Hair et al., 1979*). Taking this approach as a reference in our study, it was planned to take a sample size of at least 180 patients for nine items.

### Statistical analysis

The data were analyzed on SPSS 11.5 and AMOS 24.0 software. As descriptive statistics, mean ± standard deviation and median (minimum-maximum) were utilized for quantitative variables, and number of patients (percentage) were used for qualitative variables. For quantitative variables, Mann-Whitney U test was performed to see whether there was a statistically significant difference between categories of the qualitative variable with two categories. For quantitative variables, Kruskal Wallis H test was used to find out whether there was a statistically significant difference between categories of the qualitative variable with more than two categories since the assumptions of normality could not be met. Confirmatory factor analysis, and Spearman's rank correlation coefficient were used for construct validity, and concurrent validity, respectively. Kaiser-Meyer-Olkin (KMO) test was used to establish whether the sample size examined in the factor analysis was fit for the analysis. Bartlett's test of sphericity was performed to see whether the correlation matrix was fit for the factor analysis. Intraclass Correlation Coefficient (ICC) was utilized for the reliability of the test-retest. Cronbach's Alpha was also calculated for the reliability. Mann-Whitney U test was performed for Item Discrimination Index. Spearman's rank correlation coefficient were used to check item-total score correlations. Statistical significance level was accepted to be 0.05.

## RESULTS

### Descriptive statistics

Descriptive statistics of the Brief IPQ were provided in Table 2 for the patients who participated in the study. Significant differences were found for marital status, diabetes and periodontal diagnosis ($p = 0.004$, $p = 0.016$, and $p = 0.001$, respectively). It was determined

**Table 2 Descriptive statistics for the brief illness perception questionnaire.**

| Variables | | Total score | | |
|---|---|---|---|---|
| | | Mean ± SD | Median (Min–Max) | p-value |
| Gender | Female (n = 133) | 33.00 ± 20.22 | 31.00 (0.00–74.00) | 0.141[a] |
| | Male (n = 76) | 28.67 ± 18.63 | 27.00 (1.00–76.00) | |
| Marital status | Single (n = 84) | 26.64 ± 18.45 | 25.00 (0.00–74.00) | **0.004[a]** |
| | Married (n = 125) | 34.64 ± 19.97 | 34.00 (0.00–76.00) | |
| Educational status | Elementary school (n = 43) | 35.30 ± 22.77 | 39.00 (0.00–76.00) | 0.563[b] |
| | High school (n = 64) | 29.56 ± 19.09 | 28.50 (2.00–70.00) | |
| | University (n = 95) | 30.72 ± 18.44 | 29.00 (0.00–74.00) | |
| | Postgraduate (n = 7) | 34.29 ± 23.24 | 35.00 (5.00–64.00) | |
| Monthly income | Not working (n = 98) | 33.71 ± 20.14 | 32.50 (0.00–70.00) | 0.177[b] |
| | ≤ Minimum wage (n = 95) | 30.20 ± 19.74 | 24.00 (1.00–76.00) | |
| | > Minimum wage (n = 16) | 24.69 ± 15.35 | 23.50 (1.00–53.00) | |
| OHE experience | No (n = 97) | 32.73 ± 21.76 | 31.00 (0.00–74.00) | 0.527[a] |
| | Yes (n = 112) | 30.29 ± 17.80 | 29.00 (0.00–76.00) | |
| Smoking status | No smoking (n = 149) | 31.64 ± 19.53 | 30.00 (0.00–76.00) | 0.939[b] |
| | <10 per day | 30.75 ± 22.24 | 31.00 (2.00–71.00) | |
| | ≥10 per day | 31.04 ± 18.36 | 31.00 (1.00–63.00) | |
| Diabetes | No (n = 196) | 30.53 ± 19.51 | 29.50 (0.00–74.00) | **0.016[a]** |
| | Yes (n = 13) | 44.92 ± 18.70 | 41.00 (18.00–76.00) | |
| Hypertension | No (n = 189) | 31.17 ± 19.77 | 30.00 (0.00–76.00) | 0.541[a] |
| | Yes (n = 20) | 33.80 ± 19.57 | 33.00 (0.00–66.00) | |
| Other systemic diseases | No (n = 165) | 29.96 ± 18.74 | 29.00 (0.00–76.00) | 0.065[a] |
| | Yes (n = 44) | 36.91 ± 22.43 | 39.50 (0.00–74.00) | |
| Periodontal diagnosis | Gingivitis (n = 137) | 28.35 ± 19.49 | 25.00 (0.00–76.00) | **0.001[a]** |
| | Periodontitis (n = 72) | 37.28 ± 18.96 | 38.50 (1.00–70.00) | |

Note:
OHE, oral hygiene education; SD, standard deviation; Min, minimum; Max, maximum; a, Mann-Whitney U test; b, Kruskal Wallis H test. Significantly different values are shown in bold.

that those who were married and had diabetes had a significantly higher Brief IPQ score. Patients who had periodontitis diagnosis had a higher Brief IPQ score than patients who had gingivitis diagnosis.

## Validity

### Content validity

Content validity in the study was evaluated by 15 experts categorizing eight questions with a triple rating system as being "Essential," "Useful, but not essential," or "Not necessary". The table value of the smallest content validity ratio (CVR) for 15 experts is 0.49. CVR is calculated with the equation $CVR = [E/(N/2)] - 1$; where E: number of experts indicating "essential", and N: total number of experts. Based on the CVR values in Table 3, it was concluded that all items should be retained in the item pool since CVR values of all items are greater than 0.49.

**Table 3 Content validity ratio and content validity index values of items.**

| Items | Essential | Useful, but not essential | Not necessary | CVR | CVI |
|---|---|---|---|---|---|
| I1 | 12 | 3 | 0 | 0.800 | 0.858 |
| I2 | 14 | 1 | 0 | 0.933 | |
| I3 | 11 | 4 | 0 | 0.733 | |
| I4 | 12 | 3 | 0 | 0.800 | |
| I5 | 15 | 0 | 0 | 1.000 | |
| I6 | 14 | 1 | 0 | 0.933 | |
| I7 | 13 | 2 | 0 | 0.867 | |
| I8 | 12 | 2 | 1 | 0.800 | |

The content validity index (CVI) for the scale equals to the mean CVR across items retained in the item pool. In the present study, it was found CVI = (0.800 + 0.933 + 0.733 + … + 0.800)/8 = 0.858. As CVI = 0.858 > 0.67, the scale was concluded to be statistically significant.

### Concurrent validity

Correlation between the gold standard and the scale used in the study is investigated. If the correlation is high, it is concluded that the new scale can be used as an alternative to the gold standard. The Brief Illness Perception Questionnaire used in the present study as a new scale and Hospital Anxiety and Depression Scale used as the gold standard has two subscales which are Anxiety and Depression. Results concerning the correlation between the scales used in the study were shown in Table 4.

Correlation coefficients between the Brief IPQ total score and the HAD Anxiety, and Depression subscales were found to be 0.854, and 0.843, respectively. In addition, correlations between Brief IPQ questions and HADS subscales were significant, and correlation coefficients were found to be between 0.663 and 0.765. This results suggest that concurrent validity for the Brief IPQ was adequate (Table 4).

### Construct validity

A KMO measure of over 0.80 is expected for a good factor analysis. The KMO value of 0.922 was found in the present study, and the sample size was concluded to be adequate for the factor analysis. In addition, Bartlett's test of sphericity result was found to be significant (Chi-Square = 1,454.491, df = 28, $p < 0.001$).

A confirmatory factor analysis was used in the present study since the Turkish validity and reliability study was performed for a scale for which validity and reliability was established in its original language. Factor loadings of the items in the scale were shown in Fig. 1. Factor loadings of all items were found to between 0.773–0.879 and construct validity was established for the Brief IPQ. A Chi-square/degree of freedom ($\chi 2/df$) value below 3 is considered adequate (*Çapık, 2014*), this value was found to be 2.582 in the present study. The acceptable value for Goodness-of-fit index (GFI), Comparative fit index (CFI), and Tucker-Lewis index (TLI) is 0.9 (*Leventhal, Nerenz & Steele, 1984*). The values

**Table 4 Correlation between the brief illness perception questionnaire and hospital anxiety and depression subscales.** The Spearman correlation coefficient test was used.

| Scales | | Hospital anxiety and depression scale | |
|---|---|---|---|
| | | Anxiety | Depression |
| Consequences | Correlation coefficient | 0.763 | 0.743 |
| | p-value | <0.001 | <0.001 |
| Timeline | Correlation coefficient | 0.765 | 0.758 |
| | p-value | <0.001 | <0.001 |
| Personal control | Correlation coefficient | 0.713 | 0.663 |
| | p-value | <0.001 | <0.001 |
| Tratment control | Correlation coefficient | 0.700 | 0.685 |
| | p-value | <0.001 | <0.001 |
| Identity | Correlation coefficient | 0.718 | 0.735 |
| | p-value | <0.001 | <0.001 |
| Concern | Correlation coefficient | 0.724 | 0.731 |
| | p-value | <0.001 | <0.001 |
| Understanding | Correlation coefficient | 0.688 | 0.678 |
| | p-value | <0.001 | <0.001 |
| Emotional response | Correlation coefficient | 0.758 | 0.745 |
| | p-value | <0.001 | <0.001 |
| Brief IPQ total score | Correlation coefficient | 0.854 | 0.843 |
| | p-value | <0.001 | <0.001 |

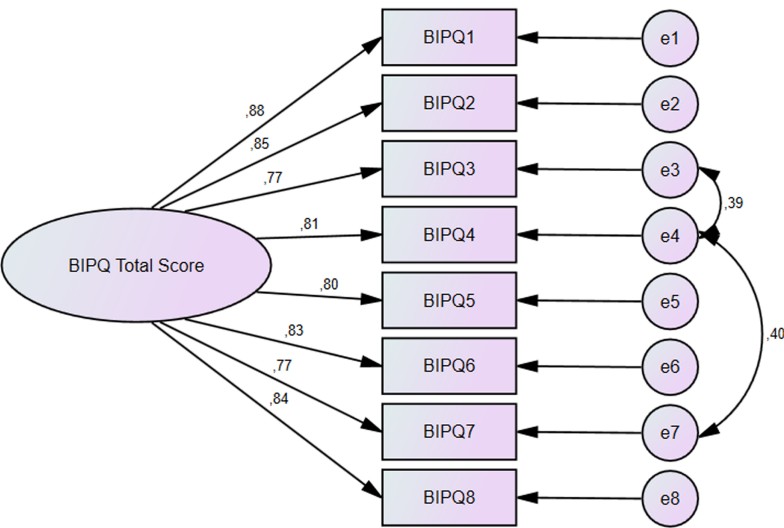

**Figure 1 Confirmatory factor analysis results for the brief illness perception questionnaire.**

found in the present study were 0.953 for GFI, 0.983 for CFI, and 0.969 for TLI. The acceptable value for RMSEA is 0.08 (*Leventhal, Nerenz & Steele, 1984*), this value was found to be 0.08 in the present study. In summary, construct validity was established in terms of the criteria used for validity.

## Reliability
### Test-retest reliability
The intraclass correlation coefficient (ICC) was used for test-retest reliability. ICC value was found to be 0.987 for the Brief IPQ and the scale concluded to be reliable based on this result.

### Internal consistency
Cronbach's alpha was calculated to be 0.944 for the Brief IPQ and the scale was concluded to have high reliability.

### Comparison of top-bottom 27% groups (item discrimination index)
For the Brief IPQ, significant difference was found between the top and bottom 27% groups ($p < 0.001$) and the scale was therefore concluded to have an adequate item distinction index.

### Item-total score correlations
The item-total score correlation is required to be greater than 0.25. Items that do not meet this requirement are recommended to be removed from the scale. The item-total score correlation coefficients for the Brief IPQ were found to be between 0.824 and 0.884, and all coefficients were statistically significant ($p < 0.001$) (Table 5).

### Examination of ceiling/floor effect in the scale
If the percentage of individuals with the lowest and highest scores that can be obtained from the scale exceeds 15%, it indicates that there is a ceiling/floor effect in the study. The lowest and highest possible scores for the IBPQ were 0 and 80, respectively. While there were 4 (1.9%) participants with a score of 0 in the study, there were no participants with a score of 80. This result showed that there was no ceiling/floor effect on the scale.

## DISCUSSION
Brief IPQ is a fast, inexpensive and useful scale that measures the patient's perception of illness. In order to use this scale in periodontal diseases, theoretical model compatibility and validity and reliability should be tested. So psychometric properties of Brief IPQ's Turkish version was evaluated among patients with periodontal diseases in the study and the scale was found valid and reliable.

It is expected that Turkish version of the Brief IPQ will play a key role in assessing patients' periodontal disease perception. The low perception of illness causes the patient to be less sensitive and less worried about the symptoms of the illness. This leads to worsening of the course of the disease and delayed intervention. A high perception of the disease makes the patient conscious and believes more in the positive effects of the treatment. The IPQ-R is accepted as the gold standard in assessing the perception of illness and is

**Table 5 Item-total item correlations for the brief illness perception questionnaire.** The Spearman correlation coefficient test was used.

| Questions | | Brief IPQ total score |
|---|---|---|
| **Consequences** | Correlation coefficient | 0.884 |
| | *p*-value | <0.001 |
| **Timeline** | Correlation coefficient | 0.864 |
| | *p*-value | <0.001 |
| **Personal control** | Correlation coefficient | 0.849 |
| | *p*-value | <0.001 |
| **Tratment control** | Correlation coefficient | 0.857 |
| | *p*-value | <0.001 |
| **Identity** | Correlation coefficient | 0.824 |
| | *p*-value | <0.001 |
| **Concern** | Correlation coefficient | 0.844 |
| | *p*-value | <0.001 |
| **Understanding** | Correlation coefficient | 0.832 |
| | *p*-value | <0.001 |
| **Emotional response** | Correlation coefficient | 0.856 |
| | *p*-value | <0.001 |

frequently used. However, the Brief IPQ was developed because this scale contains too many questions and is difficult to implement in practice.

In validity and reliability studies, content validity, concurrent validity, construct validity, test-retest reliability and internal consistency are the methods that should definitely be evaluated. In addition to these, there are additional methods that can be used. However, in most of the validity and reliability studies, only a few of these methods were considered.

Content validity was evaluated only in the study of type 2 diabetes mellitus patients and the CVI value for four expert was found to be 0.95 (*Rias et al., 2021*). In our study, content validity was calculated as a result of 15 expert evaluations and the CVI value was found to be 0.858.

In the original study of Brief IPQ, the IPQ-Revised scale was chosen for concurrent validity and the correlation coefficients were found to be between 0.24 and 0.63 as a result of the analysis (*Broadbent et al., 2006*). To assess the concurrent validity of the Brief IPQ, the correlations of the Brief IPQ with Psychological Well-being subscale of Mental Health Inventory (MHI-28) was calculated by *Bazzazian & Besharat (2010)* and correlation coefficients were found to be between 0.112 and 0.679. There are many studies in the literature that use more than one scale as the gold standard for concurrent validity and find the desired level of correlation between these scales and the Brief IPQ (*Rias et al., 2021*; *Chew et al., 2017*; *Zhang et al., 2017*). There are also studies that take the SF-36 scale as the gold standard and evaluate the concurrent validity by looking at the relationship between the sub-dimensions of this scale and the Brief IPQ (*Hallegraeff et al., 2013*; *Karimi-*

*Ghasemabad et al., 2021*). *Kuiper et al. (2022)* and *Nowicka-Sauer et al. (2016)* took the HADS as a reference scale in their study and evaluated the concurrent validity of the Brief IPQ using this scale. Similarly, in our study, HADS was taken as the gold standard, and the correlation coefficients between the anxiety and depression sub-dimensions of this scale and the Brief IPQ were found to be 0.854 and 0.843, respectively.

In the original study, factor analysis was not performed, but it was reported that a three-factor structure or a single-factor structure to be obtained by summing the scores of all questions would be appropriate (*Broadbent et al., 2006*). *Machado et al. (2019)* performed Confirmatory Factor Analysis (CFA) for construct validity and found the single-factor structure to be significant. They reported that the $\chi2/df$ value of this factor structure was 2.577, the RMSEA value was 0.053, the GFI value was 0.985, and the CFI value was 0.985 (*Machado et al., 2019*). *Bazzazian & Besharat (2010)* found the single-factor structure to be significant in the study in which they used CFA.

*Rias et al. (2021)* on the other hand, found the two-factor structure to be significant as a result of Exploratory Factor Analysis (EFA) and reported that factor loads ranged from 0.39 to 1.00. In addition, they found $\chi2/df = 2.27$, CFI = 0.96, TLI = 0.94, GFI = 0.93 and RMSEA = 0.09 of the factor structure (*Rias et al., 2021*). Similarly, there are studies that validate the two-factor construct using EFA for the construct validity of the Brief IPQ (*Zhang et al., 2017*; *Karimi-Ghasemabad et al., 2021*; *Nowicka-Sauer et al., 2016*; *Rajah et al., 2021*).

The study of *Kuiper et al. (2022)* is the only study that found the three-factor structure of the EFA result significant. In our study, a single factor structure consisting of eight items was found to be significant using CFA. While the $\chi2/df$ value of the factor structure was found to be 2.582, the GFI value was 0.953, the CFI value was 0.983, the TLI value was 0.969, and the RMSEA value was 0.08.

In this study, the ICC test was used for test-retest reliability, and the value of this test was found to be 0.987. Similarly, *Hallegraeff et al. (2013)*, *Karimi-Ghasemabad et al. (2021)* and *Rias et al. (2021)* reported that test-retest reliability was achieved by using the ICC test. In other studies investigating the validity and reliability of the Brief IPQ, the test-retest reliability was examined with the correlation test and the values were found at the desired level (*Broadbent et al., 2006*; *Bazzazian & Besharat, 2010*; *Chew et al., 2017*; *Rajah et al., 2021*).

In our study, Cronbach's alpha value was calculated for internal consistency, similar to the literature, and it was found to be 0.944. *Karimi-Ghasemabad et al. (2021)* found Cronbach's alpha 0.90 for the Brief IPQ, while *Machado et al. (2019)* found it 0.80. Similarly, Cronbach's alpha value was reported to be sufficient in other studies (*Rias et al., 2021*; *Hallegraeff et al., 2013*; *Kuiper et al., 2022*; *Nowicka-Sauer et al., 2016*; *Rajah et al., 2021*; *Saarti et al., 2016*).

The major strength of this study is that none of the validity and reliability studies of the Brief IPQ have evaluated so many criteria together. Our study is the first study to confirm the validity and reliability of the scale according to all criteria. The primary limitation of this study is its cross-sectional design. Due to the cross sectional design, only gingivitis and periodontitis patients were included in the study. Another limitation of the study is that

reassessment of the Brief IPQ after periodontal therapy was not included in the study plan. Considering the chronic nature of periodontal diseases, patients need lifelong maintenance and treatment. Therefore, patients' perceptions of illness may change after surgical and non-surgical periodontal treatments. However, this study is important in that it leads to future studies that include repeated measures of psychometric variables after periodontal treatments. It is recommended that studies with Brief IPQ may be conducted in a large population, including different periodontal diseases before and after periodontal treatment.

## CONCLUSIONS

Within the limitations of the present study, we can conclude that the Turkish version of the Brief IPQ is a valid and reliable tool to assess the ilness perception of the patients with periodontal diseases. Patients may have both symptoms and cognitive ideas about their illness. The patient's perspective on the symptoms caused by the illness may be quite different from the dental professionals. Therefore, understanding the psychological characteristics of patients may assist in designing a customized periodontal treatment plan. Brief IPQ may be used to determine the illness perception in patients with periodontal diseases because it contains few questions and is easily applicable.

### Funding
The authors received no funding for this work.

### Competing Interests
The authors declare that they have no competing interests.

### Author Contributions
- Canan Önder conceived and designed the experiments, performed the experiments, authored or reviewed drafts of the article, and approved the final draft.
- Batuhan Bakirarar conceived and designed the experiments, performed the experiments, analyzed the data, prepared figures and/or tables, authored or reviewed drafts of the article, and approved the final draft.

### Human Ethics
The following information was supplied relating to ethical approvals (*i.e.*, approving body and any reference numbers):
    Ankara University Faculty of Dentistry Clinical Studies Ethics Committee (ethical approval number: 36290600/68/2022) approved this study.

### Data Availability
    The raw data are available in the Supplemental File.

## Clinical Trial Registration

The following information was supplied regarding Clinical Trial registration:

No trial registry ID.

## Supplemental Information

Supplemental information for this article can be found online at http://dx.doi.org/10.7717/peerj.16065#supplemental-information.

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
