# Peer review of "Evaluating the Turkish validity and reliability of the Brief Illness Perception Questionnaire in periodontal diseases"

_PeerJ, doi:10.7717/peerj.16065_

## Round 0.1 · original submission · Major Revisions

This manuscript seeks to fill an important gap, the lack of studies that assess the validity and reliability of the brief IPQ in the Turkish population.

In view of the reviewers' criticisms, I think that the final quality of the manuscript could improve after a major review.

Therefore, I encourage authors to revise the manuscript according to the reviewers' criticisms and to resubmit the same

Reviewer 1 ·

Basic reporting

Dear authors,
I evaluated the article “Evaluating the Turkish validity and reliability of the Brief Illness Perception Questionnaire in periodontal diseases”.


ABSTRACT: I considered it incomplete and poorly described. Improve it!

INTRO: No study was presented in this section talking about the perception in periodontal diseases. Is it the only study?

M&M:
- It is mandatory to present the sample size calculation to respond the goal of this study. The justification used based on Hair et al., 1979 is not valid. Please, improve it; or justify that 209 patients represent a total population.

- It was presented a poor eligibility criteria.

- Suddenly, in the M&M, appears the HAD scale. It was not cited in the intro.

RESULTS
- Where is the calibration of the evaluators?
- There is lack of results. Please, see it again.

DISCUSSION: Ok!

CONCLUSION: It needs to be reviewed thoroughly.

Experimental design

There is a gap in this study using 209 patients to represent a population without a solid sample size calculation.

Validity of the findings

Extremely questionable.

Reviewer 2 ·

Basic reporting

In this study, authors aimed to assess the validity and reliability of the Brief Illness Perception Questionnaire in periodontal diseases for Turkish. Overall, the study seems methodologically sound and the statistical analyses adequate regarding the study aims and design. The strengths and limitations of the study are missing, and they must be discussed . Furthermore there is a lack of information regarding periodontal disease diagnosis, classification and characteristics.

Experimental design

There is a lack of information regarding periodontal disease diagnosis, classification and characteristics.

Validity of the findings

In this study, authors aimed to assess the validity and reliability of the Brief Illness Perception Questionnaire in periodontal diseases for Turkish. Overall, the study seems methodologically sound and the statistical analyses adequate regarding the study aims and design. The strengths and limitations of the study are missing, and they must be discussed . Furthermore there is a lack of information regarding periodontal disease diagnosis, classification and characteristics.

Additional comments

In this study, authors aimed to assess the validity and reliability of the Brief Illness Perception Questionnaire in periodontal diseases for Turkish. Overall, the study seems methodologically sound and the statistical analyses adequate regarding the study aims and design. The strengths and limitations of the study are missing, and they must be discussed . Furthermore there is a lack of information regarding periodontal disease diagnosis, classification and characteristics.

---

## Round 0.2 · accepted · Accept

The authors satisfactorily answered the questions posed in the first review. One of the reviewers who suggested rejecting the article was not contacted and due to the difficulty in recruiting other reviewers, I reviewed the second version of the article which I think is now ready to be considered for publication.

Reviewer 2 ·

Basic reporting

I am overall satisfied with the responses provided by the authors.

Experimental design

I am overall satisfied with the responses provided by the authors.

Validity of the findings

I am overall satisfied with the responses provided by the authors.

Additional comments

I am overall satisfied with the responses provided by the authors.